# Seroprevalence of Anti-SARS-CoV-2 Antibodies in Chattogram Metropolitan Area, Bangladesh

**DOI:** 10.3390/antib11040069

**Published:** 2022-11-07

**Authors:** Jahan Ara, Md. Sirazul Islam, Md. Tarek Ul Quader, Anan Das, F. M. Yasir Hasib, Mohammad Saiful Islam, Tazrina Rahman, Seemanta Das, M. A. Hassan Chowdhury, Goutam Buddha Das, Sharmin Chowdhury

**Affiliations:** 1One Health Institute, Chattogram Veterinary and Animal Sciences University, Khulshi, Chattogram 4225, Bangladesh; 2Department of Pathology and Parasitology, Faculty of Veterinary Medicine, Chattogram Veterinary and Animal Sciences University, Khulshi, Chattogram 4225, Bangladesh; 3COVID-19 Detection Laboratory, Chattogram Veterinary and Animal Sciences University, Khulshi, Chattogram 4225, Bangladesh; 4Department of Anesthesiology and ICU, Chittagong Medical College Hospital, Chattogram 4203, Bangladesh; 5Department of Infectious Diseases and Public Health, City University of Hong Kong, Hong Kong SAR, China; 6Department of Emergency and Accident, Imperial Hospital Limited, Chattogram 4202, Bangladesh; 7Department of Microbiology and Virology, Chittagong Medical College, Chattogram 4203, Bangladesh; 8Department of Medicine, Chittagong Medical College, Chattogram 4203, Bangladesh; 9Department of Animal Science and Nutrition, Chattogram Veterinary and Animal Sciences University, Khulshi, Chattogram 4225, Bangladesh

**Keywords:** seroprevalence, anti-SARS-CoV-2 antibody, antibody titer, IgG

## Abstract

Seroprevalence studies of COVID-19 are used to assess the degree of undetected transmission in the community and different groups such as health care workers (HCWs) are deemed vulnerable due to their workplace hazards. The present study estimated the seroprevalence and quantified the titer of anti-SARS-CoV-2 antibody (IgG) and its association with different factors. This cross-sectional study observed HCWs, in indoor and outdoor patients (non-COVID-19) and garment workers in the Chattogram metropolitan area (CMA, N = 748) from six hospitals and two garment factories. Qualitative and quantitative ELISA were used to identify and quantify antibodies (IgG) in the serum samples. Descriptive, univariable, and multivariable statistical analysis were performed. Overall seroprevalence and among HCWs, in indoor and outdoor patients, and garment workers were 66.99% (95% CI: 63.40–70.40%), 68.99% (95% CI: 63.8–73.7%), 81.37% (95% CI: 74.7–86.7%), and 50.56% (95% CI: 43.5–57.5%), respectively. Seroprevalence and mean titer was 44.47% (95% CI: 38.6–50.4%) and 53.71 DU/mL in the non-vaccinated population, respectively, while it was higher in the population who received a first dose (61.66%, 95% CI: 54.8–68.0%, 159.08 DU/mL) and both doses (100%, 95% CI: 98.4–100%, 255.46 DU/mL). This study emphasizes the role of vaccine in antibody production; the second dose of vaccine significantly increased the seroprevalence and titer and both were low in natural infection.

## 1. Introduction

Chattogram, the port city of Bangladesh, is classified as a high-risk zone for SARS-CoV-2 contact transmission and is one of the most crowded economic and trading centers [1]. On 3 April 2020, Chattogram city witnessed its first coronavirus disease 2019 (COVID-19) positive case [2], followed by the first death on 9 April [3]. The disease can manifest itself in various ways, from asymptomatic and minor upper respiratory symptoms to severe pneumonia and acute respiratory distress syndrome [4]. While nucleic acid amplification such as polymerase chain reaction (PCR) is the gold standard for diagnosing acute SARS-CoV-2 infection and is widely recommended, the antibody-based approach improves diagnosis accuracy by capturing asymptomatic testing and recovered infections [5].

During an infectious disease outbreak, seroprevalence investigations are crucial in revealing undetected infection in the population and preventing post-pandemic reappearance [6]. Determining the actual burden of infection is also vital for epidemic forecasting and response planning. Seroprevalence studies are potent in identifying the number of undiagnosed missing cases with mild or no symptoms or who cannot undergo testing, which may contribute significantly to the transmission [7,8,9,10,11]. Furthermore, seroprevalence studies estimate the susceptible population in a community. A current investigation discovered that up to 23% of the patients diagnosed with COVID-19 from December 2020 to February 2021 in Bangladesh were asymptomatic [12]. Thus, antibody testing could be crucial to determine the actual SARS-CoV-2 exposure rates since PCR only identifies the viral nucleic acid in individuals with existing symptoms [13].

According to numerous research, seropositivity fluctuates considerably depending on parameters such as location and time [7,14]. Antibody titers reach their peak one month after the onset of symptoms, and their levels are directly proportional to the severity of the illness [15]. Titers continue to fall after that, with IgM and IgA titers falling fast and IgG titers falling more slowly [16]. However, a greater understanding of antibody responses to SARS-CoV-2 after natural infection might aid in the development of more successful vaccination strategies in the future. Bangladesh started administering COVID-19 vaccinations on 27 January 2021, and mass immunization commenced on 7 February 2021 [17,18]. As of 21 December 2021, 50.27% of the target population had received the first dose, and 34.60% received the second dose [19]. Bangladesh has already started administering third doses to senior persons aged 60 and up, people with comorbidities, and frontline workers. [20]. According to a web-based anonymous cross-sectional survey conducted among the general Bangladeshi population between 30 January 2002 and 6 February 2002, 61.16% of respondents were inclined to accept/take the COVID-19 vaccine [21]. However, vaccination coverage and seroprevalence among the general public must be investigated nationwide to understand the herd immunity.

In the COVID-19 pandemic, HCWs are facing immense challenges worldwide. Occupational exposures among HCWs have been documented in numerous nations as worrying [22]. Likewise, COVID-19 has had a significant impact on the health care system of Bangladesh. According to the latest data from the Bangladesh Medical Association, between 8 March 2020 and 11 November 2021, 9455 HCWs including physicians, nurses, and other staff were infected with COVID-19 and 188 doctors died as a result [23]. Front liners directly involved in diagnosing, treating, and caring for COVID-19 patients are at risk of physical and psychological distress [24,25,26,27,28,29]. Similarly, workers in the garment industry confront different problems in the workplace worldwide. According to the Bangladesh Garment Manufacturers and Exporters Association (BGMEA), 4500 garment companies employ over 4.5 million people or nearly 2.5 percent of the country’s entire population [30]. The bulk of the industries operate with limited space, making it challenging to enforce physical distancing norms [31]. SARS-CoV-2 transmission might be exacerbated by crowded workplaces, transportation, and the lack of physical distancing [32]. Hence, it is necessary to put in place measures including risk management in the workplace, vulnerable employee care, the development of an occupational surveillance system, and vaccination policy administration to address the COVID-19 issues [33,34]. Thus, knowing the true seroprevalence both in the risk groups and community might assist in planning interventions efficiently. 

In this study, we reported population-based SARS-CoV-2 seropositivity among HCWs, indoor and outdoor patients of various government and private hospitals, and garment workers in the CMA, as determined by enzyme-linked immunosorbent assay (ELISA). Moreover, we measured the antibody titer, and both outcomes (seropositivity and antibody titer) were tested to learn the association between different factors.

## 2. Materials and Methods

### 2.1. Study Design and Setting

From February to September 2021, we conducted a cross-sectional population-based study among HCWs (e.g., doctors, nurses, hospital staff, ward boy, and cleaner), garment workers, and indoor and outdoor patients (non-COVID-19) of six government and private hospitals each, and two garment factories in the CMA. All hospitals belonging to the study area were stratified according to their affiliation status: government and private. From each stratum, six hospitals were randomly selected. Sample size was calculated considering the following parameters: 0.65 proportion, 5% margin of error, 95% confidence limit. and design effect 2. Each organization’s human resources department provided a list of personnel. Following a simple random sampling technique, samples were collected from a total of 748 respondents.

We interviewed participants to collect information after receiving written consent. Answering a questionnaire and taking blood to test SARS-CoV-2 antibodies were part of the study procedure. Our study followed a World Health Organization protocol for population-level COVID-19 antibody testing [35]. The questionnaire included sociodemographic details and factors hypothesized to be associated with seropositivity. Participants were included in the study based on several inclusion criteria.

#### 2.1.1. Inclusion Criteria

Asymptomatic

Only an asymptomatic group was included to ensure the presence of antibodies. Participants had no COVID-19 related clinical signs (e.g., fever, coughing, runny nose, sore throat, dyspnea, shortness of breath, aches and pain at the time of sample collection).

##### In Case of Having Past Confirmed COVID-19 Status (by Rt PCR)

Participants who had already passed at least 28 days after a negative Rt-PCR test;Participants who did not take a repeated test to ensure negativity had passed at least 42 days after the first COVID-19 test.

Furthermore, persons under 18 were excluded, as were those with an incomplete questionnaire.

### 2.2. Baseline Blood Collection and Processing

Heparinized blood specimens (6 mL) were collected and transported to the clinical pathology laboratory (CPL) of Chattogram Veterinary and Animal Sciences University (CVASU) within three hours of collection. The serum was separated to evaluate the IgG antibody and kept at −20 °C until serological investigation.

### 2.3. Serological Test Examination

Antibodies were determined by a commercial qualitative assay using a COVID-19 IgG ELISA test (Beijing Kewei Clinical Diagnostic Reagent Inc., Beijing, China; Ref: 601340) as per the manufacturer’s instructions. The assay is an enzyme-linked immunoassay (ELISA) that detects IgG against the SARS-CoV-2. An index (Absorbance/Cutt-off) of <1 was interpreted as negative, 0.9 to 1.1 as borderline (retesting of these specimens in duplicate was conducted to confirm the results), and ≥1 index as positive. As per the manufacturer, the sensitivity and specificity of the assay for IgG are 93.8% and 97.3%, respectively. Positive and negative controls were included in all assay batches. Repeated testing using the same specimen yielded the same interpretation.

The concentration of IgG antibodies was determined by SARS-CoV-2 S1-RBD IgG (DiaSino^®^ Laboratories Co. Ltd., Zhengzhou, China, Ref: DS207704), which is based on enzyme-linked immunoassay for the quantitative detection of IgG antibodies. The assay’s sensitivity and specificity for IgG quantification, according to the manufacturer, are 98.41% and 98.02%, respectively. Quantitative results were calculated as a ratio of the extinction of the control or tested specimen over the extinction of the calibrator. Results were reported in standardized units for the quantitative kits that included six calibrators to quantify the antibody concentration (i.e., DiaSino units/mL). A value of <10 DU/mL was considered negative, and values >10 DU/mL were positive.

### 2.4. Data Management

The linearity of the quantitative variables was evaluated by categorizing them into four categories using the quartiles as cut-off values. Logistic regression analysis was conducted on the categorized variables, and parameter estimates were observed for an increasing or decreasing trend. In the case of linear increase or decrease in the parameter estimates, linearity in the quantitative variable was assumed and used without modification. In the case of nonlinearity, a quartile was used to categorize it. However, some quantitative variables were categorized considering the research interest. For instance, the number of days between the first dose of vaccine and quantification of antibody titer was categorized as ‘after one month’ and ‘after two months’ and between the second dose of vaccine and quantification of the antibody titer was categorized as ‘after two months’, ‘after four months’, and ‘after six months’. The number of days between the vaccination and the antibody titer was achieved from the date of vaccination and sample collection. The prevalence estimates were adjusted with the test kit performance (sensitivity and specificity), and the adjusted prevalence was denoted as the true prevalence.

### 2.5. Data Analysis

In the study period, a total of 748 qualitative and quantitative test results were included in the analysis. To evaluate the correlation and collinearity in the categorical and quantitative variables, Cramer’s V test, Spearman correlation coefficient, Chi-square test, t-test, or ANOVA, where appropriate, was used. Variables with a significant association or a Spearman correlation coefficient above 0.4 were regarded as correlated. The effects of different potential explanatory variables on the binary outcome—presence/absence of anti-SARS-CoV-2 antibody—was evaluated using univariable and followed by multivariable logistic regression models. To select the final multivariable model, all variables with a significant *p*-value in the univariable models were included in a model and a manually conducted backward selection strategy was followed by deleting one variable at a time with the highest *p*-value. Interactions between all explanatory variables (two ways) were evaluated in the final model. The effect of variables on the mean titer of the antibody was assessed by t-test and one way ANOVA. *p*-values < 0.05 were considered as significant throughout the analysis. STATA-IC 13 (StataCorp, College Station, TX, USA) and GraphPad Prism 7.00 for Windows (GraphPad Software, La Jolla, CA, USA) were used for statistical analyses and visualization.

### 2.6. Ethical Approval and Informed Consent

Institutional ethical approval was taken from the authorized committee of Chattogram Veterinary and Animal Sciences University (CVASU), Bangladesh [CVASU/Dir(R&E) EC/2020/212(1)].

## 3. Results

### 3.1. Seroprevalence of SARS-CoV-2 Infection

SARS-CoV-2 IgG antibodies were detected in 498 (66.99%) of 748 individuals (Table 1). The prevalence of anti-SARS-CoV-2 antibody (IgG) in different donor types along with vaccination percentage is shown in Figure 1.

### 3.2. Characteristics of Study Participants

From February to September 2021, we enrolled 748 CMA service providers (362 HCWs, 205 garments workers, 179 indoor/outdoor patients). Among them, 27.48% were garment workers, 150 (20.11%) hospital staff, 145 (19.44%) doctors, 148 (19.84%) outdoor patients, 67 (8.98%) nurses, and 31 (4.16%) indoor patients. The majority (*n* = 507; 67.96%) were males. In the total population, 292 (39.14%) did not receive any COVID-19 vaccine, 223 (29.89%) received the first dose of vaccine, and 231 (30.97%) received both doses of the vaccine. The responses regarding contact with confirmed COVID-19 cases were: yes (342; 47.17%), no (307; 42.34%), and unknown (76; 10.48%). One hundred and ninety-seven (32.35%) participants had pre-existing medical conditions (Table 2).

### 3.3. SARS-CoV-2 Antibody Titer

In/outpatients had the highest mean titer of 197.18 DU/mL, followed by HCWs (163.30 DU/mL) and garment workers (77.05 DU/mL) (*p* < 0.001). The level (mean) of IgG-spike antibodies in recipients of both doses of vaccine was higher (255.46 DU/mL) than in those who received one (159.08 DU/mL) or no doses (53.71 DU/mL) of the vaccine (*p* < 0.001). When the participants who had contact with confirmed cases had a mean titer of 170.89 DU/mL, not known had a titer of 160.05 DU/mL, and in the case of noncontact, 116.45 DU/mL (*p* < 0.001). The mean titer of different age groups was statistically significant; nevertheless, we removed this variable from further analysis to minimize the bias due to the vaccination strategy followed in Bangladesh (priority given to aged); details in Table 3. The changes in mean titer of the IgG antibody across different time intervals of intervention (one and both doses of vaccination) is illustrated in Figure 2.

### 3.4. Risk Factor Analysis

#### 3.4.1. Univariable Analysis (χ^2^ Test, Logistic Regression) to Evaluate the Association of Different Variables with the Seroprevalence of Anti-SARS-CoV-2 Antibody

Indoor/outdoor patients amongst the different donor groups had a positivity rate of 81.37% (144 of 179) compared to 68.99% (248 of 362) in the HCWs and 50.56% in the garment workers (104 of 205); the difference was statistically significant (*p* < 0.001). Both doses of vaccine receivers showed significantly (*p* < 0.001) higher seropositivity than one dose or no vaccine receivers. Similarly, contact with confirmed COVID-19 cases showed a higher odd of being seropositive compared to noncontact (*p* = 0.01) [OR = 1.59] (Table 4).

#### 3.4.2. Multivariable Analysis (Logistic Regression) to Determine the Potential Factors Associated with SARS-CoV-2 Antibody-Positive Status in the Study Area

The multivariable logistic regression model identified two potential factors that might influence the seropositivity of SARS-CoV-2 antibodies in the studied population. The chance of being seropositive was 2.22 times higher in indoor/outdoor patients (*p* = 0.002) and 1.69 times for garment workers than HCWs (*p* = 0.01). Furthermore, both doses of vaccine receivers had a higher chance of being positive (OR = 174.02) than the one dose (OR = 2.34) or the none dose receivers, and the difference was statistically significant (*p* < 0.001) (Table 5).

## 4. Discussion

The overall adjusted seroprevalence estimate of SARS-CoV-2 antibodies was 66.99% (95% CI: 63.40–70.4%) in the CMA in this research, which was slightly higher than a previous finding (64.1%) using an immunoassay test to detect antibodies in the Sitakunda sub-district (Chattogram district) of Bangladesh from March to June 2021 [36]. Another study conducted by icddr’b between October 2020 and February 2021 found a lower (55%) estimate in Chattogram than ours. During the same study period, however, the adjusted seroprevalence in Dhaka, the capital of Bangladesh, was 71% [37]. Thus, based on several investigations, it can be assumed that seropositivity in Chattogram has been progressively increasing over time. The prevalence might have increased due to either high infection levels or a positive response to the national immunization campaign in its early phases [38]. According to the findings, 68.99% of HCWs and 81.37% indoor/outdoor patients were seropositive. Indoor and outdoor patients were more likely than health professionals to be seropositive, possibly due to the combined effect of a lack of awareness and knowledge about COVID-19 among some of them and the effect of vaccination as they might be composed of a mixed population of lower to upper socio-economic status with different educational levels. Tripathi et al. (2020) reported that HCWs were more educated with regard to the COVID-19 symptoms, incubation time, problems in high-risk patients, and had greater access to therapy than other residents (non HCWs) [39]. In Navi Mumbai in May 2021, serosurveillance of anti-SARS-CoV-2 antibodies among essential workers revealed that police personnel had a 72% seropositivity rate, whereas HCWs had a 48% positivity rate [40]. Moreover, we observed that, among the garment workers, just under 20% received vaccines and just above 50% were seropositive, which might have majorly been achieved from natural infections (Figure 1). It might indicate their lack of awareness about disease transmission and vaccination.

We found that the IgG antibody was produced in 61.66% of the participants who received the first dose of COVID-19 vaccination. This number increased to 100% among individuals who received a second dose. In a study by Bayram et al. (2021), HCWs’ seropositivity rates after the first and second doses of CoronaVac vaccination were found to be 77.8% and 99.6%, respectively [41]. Subsequently, when we quantified the antibody titer, we observed that it was higher in those who received a second dose than in those who received just the first. Detection of highly avid anti-S1/-RBD IgG, independent of the causal mechanism, is seen as a very positive indication and indicator of enhanced humoral immunity [42].

Human coronavirus infection may not always result in long-lasting antibody responses, with antibody titers dropping over time [43]. The waning of antibody responses is an essential element to consider when developing a coronavirus vaccine [44]. Our study showed that by the second month following the initial dose, the mean IgG titer in the body had dropped by nearly 25%. However, the antibody’s propensity to deteriorate with time was noteworthy. This study revealed that the available mean antibody titers that remained after two months of receiving the second dose had dropped by roughly 21% by the fourth month, and within the sixth month, the mean antibody titer was 147.09 DU/mL. Therefore, it can be assumed that the body still retained considerable antibodies against COVID-19 six months after receiving the second dose vaccine, though the threshold level to prevent the virus is not known.

The underreporting of SARS-CoV-2 infection cases makes it difficult to assess the actual infection burden. Limited testing, flaws in the reporting infrastructure, and a substantial proportion of asymptomatic infections contribute to the underreporting [45]. Asymptomatic carriers spread COVID-19, but the clinical characteristics, viral dynamics, and antibody responses of these individuals are unknown [46]. According to our findings, 67.66% of the asymptomatic population was seropositive where only 29.03% of asymptomatic individuals received the first dose of COVID-19 vaccine, and 28.84% also received the second dose. According to various population-based studies, a considerable majority of seropositive people were asymptomatic or had no known encounter with a COVID-19 patient [47,48,49]. Meanwhile, the observation that asymptomatic people had lower mean IgG levels than symptomatic people back up previous findings that asymptomatic carriers have less of a humoral immune response to COVID-19 infection [47,50]. The study also revealed that people aged above 35 had a greater seroprevalence. Higher seroprevalence among adults could be associated with increased vaccination exposure. On 26 January 2022, the government began accepting registrations for the COVID-19 vaccine for persons aged 55 and up in the country [51]. In the second phase, the age limit was dropped to 40 years or more, and the vaccination of youngsters aged 12–17 has recently begun in the country [52].

The latest and more deadly SARS-CoV-2 viral strains as well as the possibility of losing immunity with time after vaccination have prompted health professionals to consider the need for boosters. Research on threshold titers giving protection and time intervals of declining immunity post-immunization for low-middle-income nations such as Bangladesh are essential before launching further booster doses. An important application of serological tests is to determine the antibody responses generated upon SARS-CoV-2 infection and vaccination [53]. The continuation of this study on those who received the second dose more than six months ago will provide an appropriate booster interval, risk population category, and overview of herd immunity. According to a recent study conducted in the greater Chattogram division, it is evident that administering the first dose (Oxford-AstraZeneca) vaccine significantly reduces the health risk during the COVID-19 infection phase [54]. Therefore, it is evident that similar research is clamoring for justifications for booster administration. Additionally, more research is required to assess the efficacy of booster doses. Government and health care professionals must adopt COVID-19 vaccine booster dose utilization guidelines that consider the risks of fading immunity, new virus strains, and prioritizing vulnerable groups.

Our study had several limitations such as the fact that we only collected samples from hospitals and the garment industry, but the results would be more representative of the community if we included other groups. We could not compare the immunological responses produced by different COVID-19 vaccine brands at the same post-vaccination interval since distinct COVID-19 vaccines were licensed and supplied to the CMA at different times. We did not reveal the type and name of COVID-19 vaccines, whereas a sufficient fraction was not covered under the vaccination program, and we were concerned about an infodemic.

## Figures and Tables

**Figure 1 antibodies-11-00069-f001:**
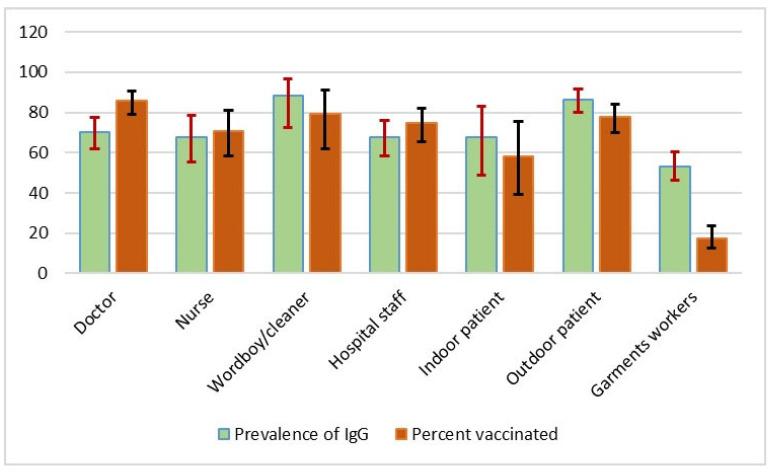
Prevalence of anti-SARS-CoV-2 antibody (IgG) in different donor types along with the vaccinated percent.

**Figure 2 antibodies-11-00069-f002:**
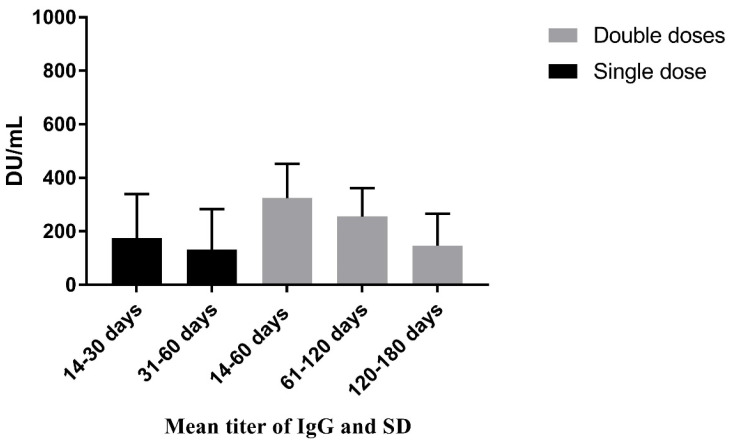
Evaluation of the time effects of the vaccines’ mean difference in the quantitative anti-SARS-CoV-2 antibody (IgG) in serum samples.

**Table 1 antibodies-11-00069-t001:** Prevalence estimation in CMA.

Anti-SARS-CoV-2 Antibody	Total Population	Unadjusted Seroprevalence, % (95% CI)	Test Performance Adjusted Seroprevalence % (95% CI)	Known Positives (RT-qPCR Positive) (%)
Present	498	66.58 (63.1–70.0)	66.99 (63.40–70.40)	91 (80.53)
Absent	250	33.42 (30.1–36.9)	32.60 (29.20–36.19)	22 (19.47)

**Table 2 antibodies-11-00069-t002:** Baseline characteristics of the study participants.

Variables	Level	Total Population	Known Positives (RT-qPCR Positive)	Asymptomatic
Donor type	Doctor	145 (19.44)	40 (35.40)	85 (16.13)
	Nurse	67 (8.98)	19 (16.81)	43 (8.16)
	Hospital staff	150 (20.11)	27 (23.89)	109 (20.68)
	Indoor patient	31 (4.16)	2 (1.77)	26 (4.93)
	Outdoor patient	148 (19.84)	21 (18.58)	109 (20.68)
	Garments worker	205 (27.48)	4 (3.54)	155 (29.41)
Gender	Male	507 (67.96)	73 (65.18)	362 (68.69)
	Female	239 (32.04)	39 (34.82)	165 (31.31)
Age (year)	19 to 29	201 (26.91)	15 (13.27)	149 (28.27)
	30 to 35	184 (24.63)	30 (26.55)	123 (23.34)
	36 to 44	180 (24.10)	34 (30.09)	123 (23.34)
	45 to 84	182 (24.36)	34 (30.09)	132 (25.05)
Vaccination	No	292 (39.14)	11 (9.82)	222 (42.13)
	Only 1st dose	223 (29.89)	38 (33.93)	153 (29.03)
	Both doses	231 (30.97)	63 (56.25)	152 (28.84)
Days passed after first dose of vaccine	14 to 30 days	45 (24.06)	8 (25.81)	30 (23.08; 16.1–31.3)
	31 to 60 days	142 (75.94)	23 (74.19)	100 (76.92)
Days passed after second dose vaccine	14 to 60 days	19 (8.26)	6 (9.38)	12 (8.00)
	61 to 120 days	86 (37.39)	20 (31.25)	60 (40.00)
	120 to 180 days	125 (54.35)	37 (59.38)	78 (52.00)
Days between PCR test and antibody test	21 to 60 days	-	17 (15.60)	-
	61 to 120 days	-	16 (14.68)	-
	121 to 180 days months	-	23 (21.10)	-
	>180 days	-	53 (48.62)	-
Contact with confirmed case	Yes	342 (47.17)	79 (71.17)	230 (45.19)
	No	307 (42.34)	17 (15.32)	232 (45.58)
	Don’t know	76 (10.48)	15 (13.51)	47 (9.23)
Family member	1 to 3	186 (26.23)	31 (29.52)	130 (25.79)
	4 to 6	443 (62.48)	64 (60.95)	321 (63.69)
	≥7	80 (11.28)	10 (9.52)	53 (10.52)
Taking immunosuppressive drugs	Yes	15 (2.13)	7 (6.42)	8 (1.63)
	No	688 (97.87)	102 (93.58)	484 (98.37)
Comorbidities	Yes	197 (32.35)	38 (37.25)	291 (68.79)
	No	412 (67.65)	64 (62.75)	132 (31.29)

**Table 3 antibodies-11-00069-t003:** Univariable analysis (*t*-test, one way ANOVA) to evaluate the mean difference in the quantity of anti-SARS-CoV-2 antibody in the serum samples.

Variable	Level	Mean Titer of IgG (DU/mL)	SD	*p*-Value
Doner type	Health worker	163.30	153.54	<0.001
	In/outpatient	197.18	147.04	
	Garment worker	77.05	115.63	
Gender	Female	140.09	151.36	0.31
	Male	151.83	148.38	
Age (year)	19 to 29	106.90	132.23	<0.001
	30 to 35	151.16	157.71	
	36 to 44	160.85	143.08	
	45 to 84	176.95	155.92	
Vaccination	No	53.71	91.16	<0.001
	Only first dose	159.08	161.05	
	Both doses	255.46	117.04	
Days passed after first dose of vaccine	31 to 60 days	131.39	152.08	0.10
	14 to 30 days	175.10	164.09	
Days passed after second dose vaccine	120 to 180 days	147.09	119.29	0.02
	61 to 120 days	255.82	106.00	
	14 to 60 days	324.42	128.42	
Asymptomatic	No	190.01	161.93	<0.001
	Yes	130.03	140.19	
Had COVID-19 confirmed status	No	191.69	142.70	0.005
	Yes	244.87	159.74	
Contact with confirmed case	No	116.45	135.21	<0.001
	Yes	170.89	154.19	
	Don’t know	160.05	158.98	
Taking immunosuppressive drugs	No	143.02	150.09	0.32
	Yes	181.38	152.08	

**Table 4 antibodies-11-00069-t004:** Univariable analysis (χ^2^ test, logistic regression) to evaluate the association of different variables with the seroprevalence of anti-SARS-CoV-2 antibody.

Variable	Level (n)	Presence of IgG	TP (95% CI of TP) **	OR	*p*-Value
Donor type	Health worker (362)	248	68.99 (63.8–73.7)	Ref.	<0.001
	Indoor/outdoor patient (179)	144	81.37 (74.7–86.7)	1.8	
	Garments worker (205)	104	50.56 (43.5–57.5)	0.47	
Gender	Female (239)	151	63.47 (56.9–69.5)	Ref.	0.15
	Male (507)	347	68.92 (64.6–72.9)	1.26	
Age (year)	19 to 29 (201)	114	56.76 (49.5–63.6)	Ref.	0.002
	30 to 35 (184)	119	65.01 (57.6–71.8)	1.39	
	36 to 44 (180)	132	73.99 (66.8–80.1)	2.09	
	45 to 84 (182)	133	73.73 (66.6–79.8)	2.07	
Vaccination	No (292)	131	44.47 (38.6–50.4)	Ref.	<0.001
	Only first dose (223)	137	61.66 (54.8–68.0)	1.95	
	Both doses (231)	229	100 (98.4–100.0)	140.72	
Days passed after first dose of vaccine	31 to 60 days (142)	79	55.64 (47.1–63.8)	Ref.	0.29
	14 to 30 days (45)	29	64.78 (49.6–77.5)	1.44	
Days passed after second dose vaccine	120 to 180 days (125)	123	99.9 (95.7–100)	-	-
	61 to 120 days (86)	86	100 (97.2–100)	-	
	14 to 60 days (19)	19	100 (84.2–100)	-	
Asymptomatic	No (220)	160	73.36 (66.9–79.03)	Ref.	0.13
	Yes (528)	355	67.66 (63.4–71.68)	0.76	
Had COVID-19 confirmed status	No (144)	119	83.65 (76.3–89.1)	Ref.	0.66
	Yes (113)	91	81.46 (72.9–87.9)	0.86	
Contact with confirmed case	No (307)	187	61.11 (55.3–66.6)	Ref.	0.01
	Yes (342)	244	71.93 (66.7–76.6)	1.59	
	Don’t know (76)	49	64.81 (53.1–75.0)	1.16	
Taking immunosuppressive drugs	No (688)	447	65.32 (61.5–68.9)	Ref.	0.20
	Yes (15)	12	80.91 (54.7–94.3)	2.15	

** TP = True prevalence.

**Table 5 antibodies-11-00069-t005:** Output from the final multivariable logistic regression model showing the adjusted effect of potential factors on the seroprevalence of the anti-SARS-CoV-2 antibody.

Variable	Level	OR	95% CI	*p*-Value
Doner type	Health worker	Ref.		
	Indoor/outdoor patient	2.22	1.33–3.68	0.002
	Garment worker	1.69	1.09–2.62	0.01
Vaccination	No	Ref.		
	Only first dose	2.34	1.56–3.50	<0.001
	Both doses	174.02	41.46–730.40	<0.001

## Data Availability

Data is contained within the article. Raw data can be obtained upon request from the director of One Health Institute of the CVASU.

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
