# Peer review of "Seroprevalence of Anti-SARS-CoV-2 Antibodies in Chattogram Metropolitan Area, Bangladesh"

_2073-4468, 2022, doi:10.3390/antib11040069_

Round 1
Reviewer 1 Report
Thank you for the article is clear and well written
Some minor cements
Why were selected garment workers instead of other professions? factories? Or related with health
Could have an influence of positivity or negativity the time after COVID-19 Contact with a confirmed case and/ or vaccination?
Is there any specific vaccine that have been used? And could be this related with seroprevalence?
Author Response
Please find the attached file of reply. Thank you

Reviewer 2 Report
In this manuscript, Ara et al., compared population-based SARS-CoV-2 seropositivity among HCWs, indoor and outdoor patients of various government and private hospitals, and garment workers of CMA. They found different seroprevalence between different populations. In addition, they concluded that second dose of vaccine significantly increased the seroprevalence. Overall, the experiments and analysis are well performed. I have a few minor comments for this manuscript.
Minor
1. Do the authors know/speculate with which variants the donors were infected?
2. Regarding the question above, the authors should include the information which variants were spread in Bangladesh (at least Dec 2020-Feb 2021).
3. Figure 1: the authors should perform statics analysis. Because prevalence of IgG from Garments workers looks significantly lower than others.
4. Figure 1: Can the authors show individual data by dot? It might be better.
5. Table 2: please add the information what the numbers in ( ) means. This should be (%).
6. Figure 2: Please clarify what the Y-axis means in the Figure?
Author Response
Please find the attached file of reply. Thank you.
